# What Matters for Model Merging at Scale?

**Prateek Yadav**$^*$                                          praty@cs.unc.edu
*The University of North Carolina at Chapel Hill*
*Google DeepMind*

**Tu Vu**                                                      tuvu@vt.edu
*Virginia Tech*
*Google DeepMind*

**Jonathan Lai**                                              jhflai@google.com
*Google DeepMind*

**Alexandra Chronopoulou**                                    alexandrachron@google.com
*Google DeepMind*

**Manaal Faruqui**$^*$                                        mfaruqui@google.com
*Google DeepMind*

**Mohit Bansal**                                              mbansal@cs.unc.edu
*The University of North Carolina at Chapel Hill*

**Tsendsuren Munkhdalai**                                     tsendsuren@google.com
*Google DeepMind*

**Reviewed on OpenReview:** *https://openreview.net/forum?id=9sbetmvNpW*

## Abstract

Model merging aims to combine multiple expert models into a more capable single model, offering benefits such as reduced storage and serving costs, improved generalization, and support for decentralized model development. Despite its promise, previous studies have primarily focused on merging a few small models. This leaves many unanswered questions about the effect of scaling model size and how it interplays with other key factors—like the base model quality and number of expert models— to affect the merged model's performance. This work systematically evaluates the utility of model merging at scale for transformer based models to examine the impact of these different factors. We experiment with merging fully fine-tuned models using four popular merging methods—`Averaging`, `Task Arithmetic`, `Dare-TIES`, and `TIES-Merging`—across model sizes ranging from 1B to 64B parameters and merging up to 8 different expert models. We evaluate the merged models on both held-in tasks, i.e., the expert's training tasks, and zero-shot generalization to unseen held-out tasks. Our wide range of experiments provide several new insights about merging transformer based models at scale and the interplay between different factors. *First*, we find that merging is more effective when experts are created from strong base models, i.e., models with good zero-shot performance, compared to pre-trained ones. *Second*, larger models perform better when merged. *Third* merging consistently improves generalization capabilities. Notably, when merging eight large expert models, the merged models often generalize better compared to the multitask trained models. *Fourth*, we can better merge more expert models when working with larger models. *Fifth*, different merging methods behave very similarly at larger scales. Overall, our findings shed light on some interesting properties of model merging while also highlighting some limitations.

---

$^*$Work done while at Google DeepMind.

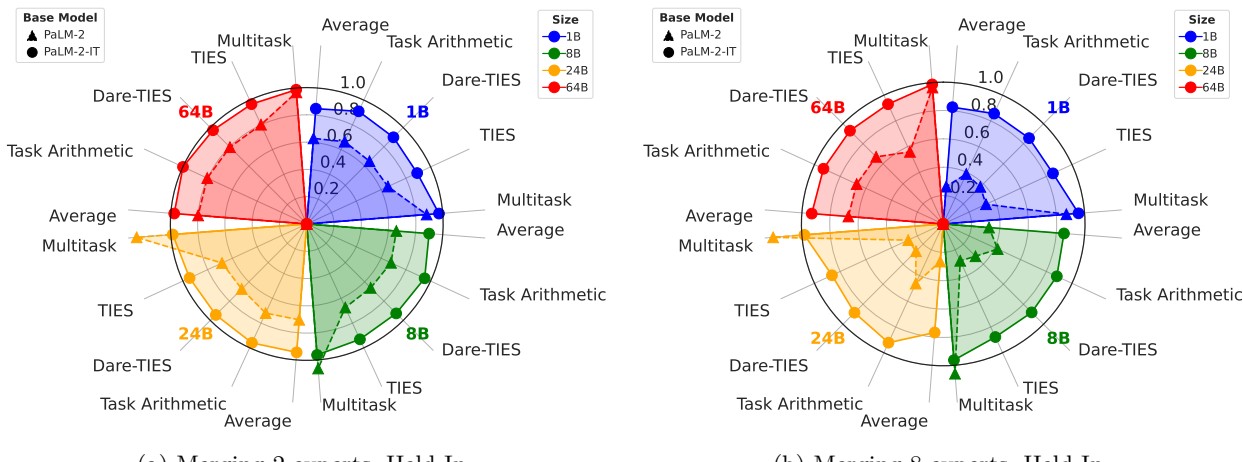

(a) Merging 2 experts, Held-In.

(b) Merging 8 experts, Held-In.

Figure 1: **Held-In performance results from our large scale model merging experiments** conducted over keys factors like **base models**, **model sizes**, **merging methods**, and **number of experts** being merged. We present results for two base models, `PaLM-2` and an instruction tuned version of it, `PaLM-2-IT`, four different models sizes (`1B`, `8B`, `24B`, `64B`), four merging methods (`Averaging`, `Task Arithmetic`, `Dare-TIES`, and `TIES-Merging`), when merging either `2` or `8` expert models. We report the performance normalized with the oracle expert's performance which is denoted by the bold black circle of radius `1`. We also present the performance of multitask baseline train on the held-in tasks. We find merging expert models created from the instruction tuned `PaLM-2-IT` model always performs better than merging `PaLM-2` based experts. Moreover, the gap between these model increase when we merge more experts. Larger experts (`64B`) merge better and show the best held-in performance.

# 1    Introduction

Model merging ([Raffel, 2021](#)) refers to the process of combining two or more *constituent* (*expert*) models to produce a new, and potentially more powerful model. The appeal of this technique is rooted in several benefits it can confer: *first*, it dramatically reduces storage and serving costs by reusing a single model across tasks; *second*, it enables compositional combination of capabilities from expert models, which can improve generalization to novel tasks; and *third*, merging supports decentralized and modular model development by allowing multiple contributors to independently build models and later combine them together.

These characteristics have led to a great deal of recent efforts in developing cost-effective model merging methods ([Matena & Raffel, 2022b](#); [Ilharco et al., 2022](#); [Jin et al., 2022](#); [Yadav et al., 2024b](#); [Yang et al., 2023](#); [Yu et al., 2024d](#); [Shah et al., 2023](#); [Tam et al., 2023](#); [Zhao et al., 2024](#)), often using simple *arithmetic operations*, such as averaging the parameters of the constituent models. However, most of these studies are limited to small-scale experiments with relatively small models (typically $< 7B$ parameters) and merging `2` or `3` experts ([Yu et al., 2024a](#);[c](#)), and mainly focus on improving benchmark performance on *held-in* tasks that the expert models were trained on ([Yu et al., 2024a](#); [Yadav et al., 2024b](#)). Despite the promises that model merging holds, the research community still lacks a comprehensive study to evaluate its effectiveness as we scale the model size. Moreover, it is not clear how scale interplays with other factors like number of expert models and base model quality to affect the merged model's held-in performance and zero-shot generalization. This is of paramount importance, as models are rapidly growing in size, and more open-weight models and datasets are becoming available,[1] driving the need for practical and scalable merging methods.

Our primary goal in this paper is to provide insight into the scalability of model merging for transformer based models. Although some studies have explored merging at the `13B` parameter scale ([Huang et al., 2024a](#); [Yu et al., 2024d](#);[b](#)), they primarily leverage increased model size and combine only `2-3` models to attain better performance on held-in tasks. As such, the interplay of factors like model size, base model quality,

---

[1]As of writing Hugging Face hosts a plethora of community-contributed resources, with `1M+` models and `200K+` datasets.

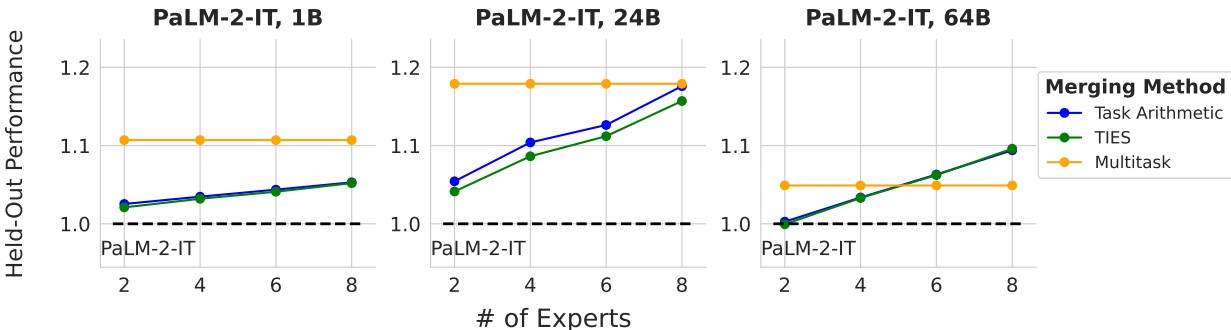

Figure 2: **Merged experts created from big and strong base models generalize better than multitask models.** We find that for strong base models as we merge more experts (x-axis, →), the merged model's generalization performance (y-axis, ↑) monotonically increases to approach and eventually surpasses multitask baseline. (yellow line). More details in Section 4.3.

number of constituent models—and their effect on both held-in and zero-shot generalization performance (*held-out*)—remains largely *unexplored.* Hence, we aim to address the following *four* research questions (RQ):

**RQ1:** What is the effect of using *pretrained* vs. *instruction-tuned* base models for creating expert models for merging?

**RQ2:** Does model merging perform *better* or *worse* as the model size increases?

**RQ3:** How does merging affect *zero-shot generalization* to held-out tasks, and how is this influenced by model size?

**RQ4:** How *many* expert models can be merged without performance loss, and how does this depend on model size?

To answer these question, we systematically evaluate the effectiveness of current *state-of-the-art* merging methods through empirical experiments. Specifically, we utilize the `PaLM-2` model (Anil et al., 2023) and its instruction-tuned variant, `PaLM-2-IT`, while scaling the model sizes up to `64B` parameters. We experiment with *four* popular merging methods, namely, `Averaging` (Wortsman et al., 2022a; Choshen et al., 2022b), `Task Arithmetic` (Ilharco et al., 2022), `TIES-Merging` (Yadav et al., 2024b), and `Dare-TIES` (Yu et al., 2024d). We conduct a series of sensitivity and ablation experiments to understand the relative importance of several factors like model size (`1B`, `8B`, `24B`, `64B` parameters), base model quality (pretrained vs. instruction-tuned), and number of constituent models (`2`, `4`, `6`, `8`) being merged. Additionally, we consider two axes of evaluation using the T0 data collection (Sanh et al., 2021a): held-in evaluation with tasks the expert models were trained on, and held-out, for zero-shot generalization to unseen tasks.

Our experiment results shed light on the promises of model merging and reveal interesting insights into the behaviors of different factors at scale. *First*, we find that the model initialization plays a crucial role in enhancing the performance of the merged model. Specifically, across all evaluation settings, using strong zero-shot instruction-tuned base models to create expert models leads to improved performance compared to using pretrained models (see §4.1). *Second*, larger models perform better when merged. This holds true regardless of the base model used (instruction-tuned or not), number of models merged, or merging method (see §4.2). *Third*, our results demonstrate that merging significantly enhances zero-shot generalization, consistently improving the ability to adapt to new tasks. *Notably*, when using strong base models as the number of merged experts increases, the merged model either matches or exceeds the performance of a strong multi-task training baseline (see §4.3). *Fourth*, larger models are better at merging a larger number of expert models (see §4.4). *Finally*, our numerous experiments identify specific settings where we expect model merging to be much more useful. From this we provide general recommendations for practitioners (see §4.7). Taken as a whole, our findings are a powerful testament to the potential of model merging at scale for creating highly generalizable language models, which we hope will spur more fundamental research into the development of practical and scalable merging methods.

## 2 Background

Model merging has emerged as a cost-effective method for developing improved models. Two common use cases of merging are: (1) combining model checkpoints from different data versions, hyperparameters, or training stages to enhance distributional robustness (Team et al., 2024; Dubey et al., 2024), and (2) combining multiple expert models trained on different datasets to leverage their complementary capabilities. In both scenarios, the expert models generally share a common architecture and a base model from which the expert models are created via fine-tuning.

This work focuses on merging specialized, fine-tuned versions (experts) of a single base model to enhance its capabilities. Each expert model is trained on distinct datasets covering different tasks, domains, and/or capabilities. We refer to the tasks/datasets used for training the expert models as "`held-in`", while those that are new and unseen are called "held-out". Our goal is to create a unified model that retains the individual expert models' capabilities on held-in tasks while improving zero-shot generalization on held-out tasks. This merging approach provides a flexible, modular method for post-training large language models, facilitating the addition of new features and capabilities to top-performing models.

### 2.1 Model Merging Methods

We denote the set of N expert tasks as $t_1, \ldots, t_N$ and the base model weights, representing the common ancestor of all expert models as $\theta_{\text{base}}$. The weights of the corresponding specialized expert models, each obtained by fully fine-tuning the base model on a specific expert task, are denoted as $\theta_1, \ldots, \theta_N$, respectively. We focus on "open vocabulary" models which utilize natural language as input and output for both classification and generation tasks, eliminating the need for task-specific classification heads making the merging process simpler. Given this, model merging methods can be defined as a function $\mathcal{M}(.)$. This function takes as input the base model, the set of N expert models, and potentially additional information, denoted by $\Phi$. This additional information may include activation statistics, Fisher matrices, or other method-specific data. The output of the function is the merged model, represented by its parameters $\theta_m$. Formally, $\theta_m = \mathcal{M}(\{\theta_i\}_{i=1}^N, \theta_{\text{base}}, \Phi)$, where $\Phi$ is method specific data.

Given our focus on studying model merging with large models, we select four merging methods based on their popularity and simplicity. We only study merging methods that can scale to tens of billions of model weight parameters and do not require any additional information to perform merging, i.e., $\Phi = \{\}$, as these techniques are efficient for even larger models. Other more complex methods that require computing fisher matrices (Matena & Raffel, 2022a), backward passes (Yang et al., 2023), or additional information like model activation (Jin et al., 2023) are skipped because of their computational complexities for large scale model merging that we focus on in this work. Next, we describe the four selected model merging methods in detail.

#### 2.1.1 Averaging

Parameter averaging (Choshen et al., 2022b; Wortsman et al., 2022a) is a well-established technique in federated learning (McMahan et al., 2017) and recent applications extend its utility to merge models for enhancing model robustness against out-of-distribution data (Wortsman et al., 2022b; Ramé et al., 2022a), refine pre-trained models (Yu et al., 2024a), develop multimodal models (Sung et al., 2023), and create multitask models by combining capabilities (Yadav et al., 2024b; Ilharco et al., 2022). Parameter averaging is achieved by taking a mean of all the expert model weights together without using the base model which can be formally described as, $\mathcal{M}(\{\theta_i\}_{i=1}^N, \theta_{\text{base}}) = \frac{1}{N}\sum_{i=1}^N \theta_i$.

#### 2.1.2 Task Arithmetic

Task Arithmetic (Ilharco et al., 2022) introduces a novel concept of "*task vectors*" for model merging. For task $t_i$, the task vector is denoted as $\tau_i = \theta_i - \theta_{\text{base}}$ which captures task-specific knowledge by quantifying the difference between the fine-tuned expert parameters ($\theta_i$) and the original base model parameters ($\theta_{\text{base}}$). A scaling hyperparameter $\lambda$ controls the contribution of the aggregated task-specific knowledge to the final model. The merged model is then constructed by linearly combining the base model parameters with a scaled sum of all task vectors. Formally, task arithmetic can be described as, $\mathcal{M}(\{\theta_i\}_{i=1}^N, \theta_{\text{base}}; \lambda) = \theta_{\text{base}} + \lambda * \sum_{i=1}^N (\theta_i - \theta_{\text{base}})$.

### 2.1.3 `TIES` **Merging**

`TIES`-Merging (Yadav et al., 2024b) identifies two main challenges with model merging: ❶ during finetuning expert models accumulate a lot of noise in the parameters, and ❷ different experts might want to change the same parameter in different directions leading to interference/conflict between the expert models. They demonstrate that both of these factors hurt model merging and propose a three steps process to remove redundant parameters, followed by resolving sign conflicts, and finally aggregating only the parameters that are not conflicting. Specifically, in `TIES` Merging they first zero out the values in each task vector that have low magnitudes to obtain the trimmed task vector $\hat{\tau}_i$ for each task. Next, they chose the aggregate sign ($\gamma_m$) for each parameter based on whether the parameter has a higher total magnitude in the positive or the negative direction across all trimmed task vector, formally, $\gamma_m = \text{sgn}(\sum_{i=1}^{N} \hat{\tau}_i)$. Next, for each parameters $p$ the models whose sign matches the aggregate sign are averaged to obtain the merged task vector. Finally, the merged model is obtained by scaling the merged task vector using a hyperparameter $\lambda$ and then added back to the base model as, $\theta_m^p = \theta_{\text{base}} + \lambda * \frac{1}{|\mathcal{A}^p|} \sum_{i \in \mathcal{A}^p} \hat{\tau}_i^p$, where $\mathcal{A}^p = \{i \in [N] \mid \hat{\gamma}_i^p = \gamma_m^p\}$.

### 2.1.4 Dare Merging

Dare (Yu et al., 2024a) extends the idea of `TIES` merging by proposing to use a dropout-like pruning stage to remove noise before merging. Specifically, a Bernoulli mask $M_i$ with drop probability $p$ is applied to each task vector to obtain the pruned task vector $\hat{\tau}_i = (1 - M_i) \odot \tau_i/(1 - p)$. This stochastic process randomly zeroes out elements within the task vector while preserving its expected value. These pruned task vectors are then used along with either `TIES` Merging or Task Arithmetic. Due to the popularity of the Dare variant that uses `TIES` Merging, we use that to represent the Dare method and call it *Dare-TIES*.

### 2.2 Challenges/Limitations

Model Merging has been utilized at a growing rate in practice as it has recently been applied to building modern language models like `Llama-3` (Dubey et al., 2024) and `Gemma-2` (Team et al., 2024). However, most formal studies on model merging have been performed with relatively small models. There are a few studies that look at larger models with `7B` and `13B` parameters. However, those studies mostly focus on merging `2-3` models to improve benchmark numbers as opposed to better understanding how the size of the model affects the model merging process and the resultant model. To motivate our work, we present some of the limitations of the existing studies and highlight their difference with our work.

**Most Studies on Small Models ($<$ `7B` parameters):**  Most existing model merging papers rarely use large models ($>$ `7B`). For example past works (He et al., 2024; Ortiz-Jimenez et al., 2024; Jang et al., 2024), including methods like ModelSoup (Wortsman et al., 2022a), Task Arithmetic (Ilharco et al., 2023) and `TIES`-Merging (Yadav et al., 2024b), Ada-Merging (Yang et al., 2023), MatS (Tam et al., 2024) perform experiments with model families like CLIP (Radford et al., 2021), ViT (Dosovitskiy et al., 2021), T5 (Raffel et al., 2020a), DeBERTa (He et al., 2021), Roberta (Liu et al., 2019), BERT (Devlin et al., 2018) with less than `1B` parameters. Hence, it is unclear how well model merging works for large models, what factors play an important role, the effect of model size, number of tasks being merged, and its effect on both held-in performance and generalization of the model. Some studies hypothesize that bigger models perform better when merged, however there are no concrete large scale studies to thoroughly assess such claims at large scale.

**Model Merging Studies with Large Models are Shallow:**  Some recent works like DARE (Yu et al., 2024a), WIDEN (Yu et al., 2024c), Chat-Vector  (Huang et al., 2024b) demonstrate merging results for larger models with up to `13B` parameters, however these studies have a few limitations: ❶ They primarily focus on using model merging to improve model quality and hence their experiments do not provide concrete insights on how model size interplays with merging, ❷ They only merge a maximum of two or three models at once, ❸ They primarily focus on held-in tasks and do not provide any insights on the effect of merging on a model's generalization abilities. Other works like RewardSoup (Rame et al., 2024), WARM (Rame et al., 2024), FuseLLM (Wan et al., 2024) also work with $\sim$ `7B` sized models and focus on specific applications of model merging without providing any deeper insight about how merging performance changes for large models.

**Varied Evaluation Setups:** Most previous works rarely share their experimental setup where both the expert datasets and the objective vary. For example, Task Arithmetic (Ilharco et al., 2023), `TIES` (Yadav et al., 2024b), MaTS (Tam et al., 2024) uses GLUE tasks (Wang et al., 2018), Vision tasks, T0 held-out, and T0 held-in (Sanh et al., 2021b) tasks respectively. Moreover, different works evaluate for different use cases like intermediate task training in Fisher merging (Matena & Raffel, 2022a), robustness in modelsoups (Wortsman et al., 2022a), and held-in performance for Dare (Yu et al., 2024a), both held-in and held-out performance in `TIES` Merging (Yadav et al., 2024b). Given our focus on combining model capabilities in the post training phase, we focus on evaluating on both held-in tasks and generalization to unseen held-out tasks.

## 3 Large Scale Evaluation of Model Merging

In this work, we address the limitations mentioned above by systematically understanding the effect of various factors like model size, base model quality, merging method, and the number of models being merged on both the held-in and generalization performance of the merged model. Next, we describe our experimental design.

**Data:** Sanh et al. (2021a) found that explicit multitask training of T5 (Raffel et al., 2020b) on a collection of prompted datasets produces a model with strong zero-shot performance on unseen tasks. This has become a common experimental setting for benchmarking zero-shot generalization (e.g. (Longpre et al., 2023; Jang et al., 2023; Zhou et al., 2022; Chung et al., 2024; Muqeeth et al., 2024). Hence, we adopt the experimental setting from the T0 mixture (Sanh et al., 2021a) which contains 8 held-in and 4 held-out task categories. For each of these categories there are multiple datasets in the T0 mixture (Sanh et al., 2021b) and hence to reduce evaluation costs, we select 2 datasets from each category based on the popularity and the train dataset size. Specifically, the 8 held-in task categories (with a total of 16 datasets) include Multiple-choice QA, Extractive QA, Closed-Book QA, Sentiment Analysis, Topic Classification, Structure-to-text, Summarization, and Paraphrase Identification. Similary, the 4 held-out task categories (with a total of 7 datasets) are Sentence Completion, Natural Language Inference, Coreference Resolution, and Word Sense Disambiguation. For more details see Section A.

**Expert Model Creation:** Recognizing the significance of post-training for LLMs where models are typically fully fine-tuned, we perform full fine-tuning to create our expert models to better mimic the post-training setting. Moreover, in post-training phases it is common to first perform Instruction Tuning (IT) on the model before moving on to other steps. Hence, we examine the effect of using strong instruction-tuned base models on the process and outcome of model merging. Given this, we utilize the transformer based `PaLM-2` models (Anil et al., 2023) with sizes `1B`, `8B`, `24B`, and `64B` as our base models ($\theta_{\text{base}}$). To obtain the instruction tuned base model, we further fine-tuned the `PaLM-2` models on the `FLAN-v2` dataset (Longpre et al., 2023) while excluding the T0-mixture tasks (Sanh et al., 2021a). These instruction-tuned variants are denoted as `PaLM-2-IT`. For each of the 2 base model types (non-IT vs IT) and 4 model sizes, we perform full fine-tuning on the 8 held-in task categories resulting 64 specialized experts models which are then used further in our experiments. Comprehensive details regarding hyper parameters and computational requirements are provided in Appendix B.

**Experimental Setting:** Given our collection of expert models, for each merging experiment we select a subset of expert models which we call the *constituent models*. We create a large merging experiment grid with 2 base models (`PaLM-2` and `PaLM-2-IT`), 4 model sizes (`1B`, `8B`, `24B`, `64B`), 4 Merging methods (Averaging, Task Arithmetic, Dare-TIES, and `TIES`), the number of constituent models (2, 4, 6, 8), and 3 seeds to randomly select the constituent tasks for the experiment resulting in a total of 384 merging experiments. These seeds are shared across different experimental settings and control the different combinations of models that are selected for merging. They ensure that the same tasks are selected across base models, model sizes and merging methods to ensure fair comparison. For example, in an experiment where we merged 2 expert models, derived from the 64B `PaLM-2` base model with the constituent models being MCQ and Summarization experts while the same experiment with a different seed resulted in Closed Book QA and Sentiment Analysis experts as the constituent models.

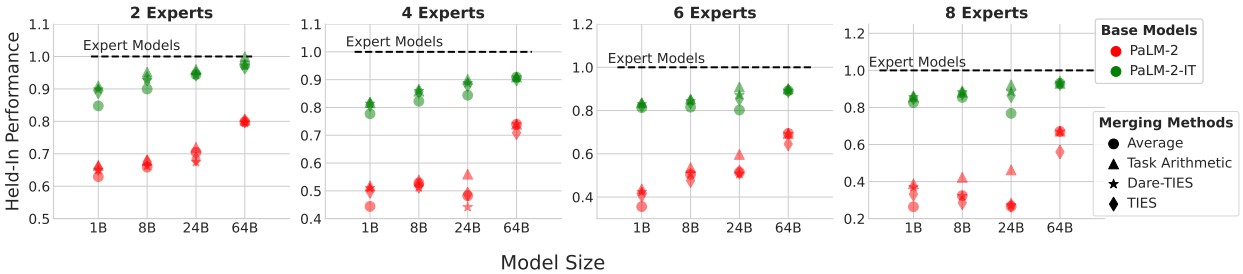

Figure 3: **Instruction-tuned models perform better when merged.** `PaLM-2-IT` (•) consistently outperforms `PaLM-2` (•) as shown by the huge gap between the green point (•) being higher than red points (•), across various merging methods, model sizes, and numbers of constituent models, indicating that stronger instruction-tuned base models enhance the performance of merged models. The dashed lines denoted the performance of the experts trained on the held-in tasks as defined in § 3. For more details see Section 4.1.

**Evaluation:** For each of the experiments above, we assess the merged model's performance by evaluating it on both the held-in tasks – i.e., the training tasks of the constituent expert models – and all 4 held-out task categories. For example, if the constituent models are MCQ and Summarization experts, then for held-in tasks we evaluate on the MCQ datasets (DREAM and Cosmos QA) and Summarization datasets (CNN Daily Mail and XSum) resulting a total of 4 held-in evaluation datasets. Moreover, all merging experiments are also evaluated on the 4 held-out tasks categories consisting of 7 datasets listed in Appendix A. There we perform approximately ∼ 9000 model evaluations across all of our experiments.

**Metric:** Given that different datasets use different metrics, we normalize the performance metrics to make them unitless so that they can be aggregated. Similar to Task Arithmetic (Ilharco et al., 2022) and TIES-Merging (Yadav et al., 2024b), for held-in tasks, the merged model's performance is normalized against the corresponding task expert model's performance. However, for held-out tasks, the normalization was performed relative to the base model's performance.

For example, say we have two held-in tasks $t_1$, $t_2$ and two held-out tasks $t_2$ and $t_4$. Now we train an expert model on $E_1$ and $E_2$ on the tasks $t_1$, $t_2$ respectively. Next, we merge experts $E_1$ and $E_2$ to obtain the merged model, $M$. Now, to evaluated $M$ on the held-in tasks, we report the average normalized performance on its held-in tasks ($t_1$ and $t_2$) as the average of $\frac{acc(M,t_1)}{acc(E_1,t_1)}$ and $\frac{acc(M,t_2)}{acc(E_2,t_2)}$, where $acc(E,t)$ means the accuracy of expert model $E$ on task $t$. For held-out tasks, the denominator is the performance of the model from which the experts are created. Concretely, the held-out performance is the average of $\frac{acc(M,t_3)}{acc(base,t_3)} + \frac{acc(M,t_4)}{acc(base,t_4)}$, where *base* is the model from which experts are created.

We denote this metric as *normalized performance* throughout the paper. Importantly, we want to emphasise that this metric is relative, with a value of 1 indicating performance comparable to the reference model. Hence, for held-in tasks a value of 1 means performance similar to the domain expert model while for held-out tasks it means performance is similar to the base model. We mark this line in most of our figures and specify the models that are used for normalization. Finally, to generate aggregated results, we compute the mean of normalized performance across all datasets within each category, then across all categories and then over the three seeds.

## 4 Experimental Results

In this section, we explore the interplay between model size and key factors such as base model quality, merging method, and the number of constituent (expert) model, along with their effect on both held-in and zero-shot generalization (held-out) performance. Our findings are: ❶ Merging is more effective when the constituent models are derived from instruction-tuned base models rather than pretrained ones (see §4.1); ❷ Larger models perform better when merged (§4.2); ❸ Merging significantly improves zero-shot generalization, with instruction-tuned models benefiting from increased constituent models, and larger model

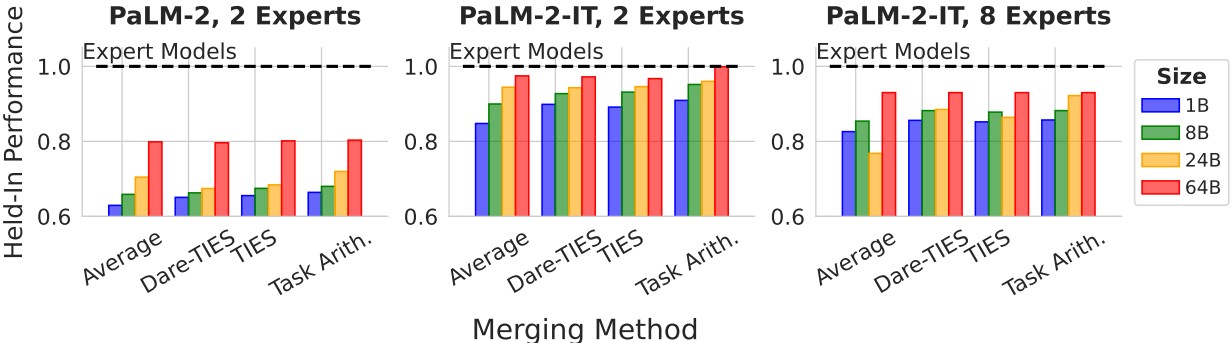

Figure 4: **Bigger models merge better.** On Held-In evaluations, we find that bigger models always perform better compared to smaller models, barring a few outliers. We find that large instruction tuned models like `64B PaLM-2-IT` perform best when merged. For more details see Section 4.2.

sizes allowing the merged model to match or exceed multi-task training (§4.3); ❹ We can merge more models effectively when using larger models (§4.4); and ❺ Different merging methods perform similarly when applied to large-scale instruction-tuned models. Below, we outline the experimental setup and discuss these findings in detail. *Our full results along with the standard deviations can be found in Appendix C and Table 2, 3,4,5.*

## 4.1 Instruction-Tuned Models Facilitate Perform Better When Merged

**Experimental Setup:** Prior works suggests a connection between good zero-shot models and effective model merging. Wortsman et al. (2022a) demonstrate that averaging strong zero-shot models improves out-of-distribution robustness. Ortiz-Jimenez et al. (2024) indicate that effective pretraining allows weight disentanglement, and thus enhances merging. Yadav et al. (2024b); Ilharco et al. (2023) propose that instruction tuned base models could aid in model merging, though this hypothesis remains largely untested.

To assess how base model quality affects the held-in performance of merged models, we perform merging experiments with fully fine-tuned experts from `PaLM-2` and `PaLM-2-IT`. We vary model sizes in $\{1B, 8B, 24B, 64B\}$ and the number of constituent models in $\{2, 4, 6, 8\}$. Held-in performance is measured over three trials in which different combinations of models are selected in order to minimize the impact of selected expert models and their data distributions on performance trends. A consistent seed is used to select the tasks across different base models, model sizes, and merging methods to ensure fair task comparisons. We evaluate four merging methods: averaging, task arithmetic, `TIES`, and Dare-TIES, and also compare against the performance of task-specific expert models.

**Findings:** Our results, presented in Figure 3, indicate that `PaLM-2-IT` models denoted by green color (•), consistently outperforms `PaLM-2` models (•) across various merging methods (•, ▲, ♦, ⋆), model sizes (x-axis →), and numbers of constituent models (subplots). This supports our hypothesis that for transformer based LLMs stronger instruction-tuned base models enhance the performance of merged models. Similar to the findings of Ortiz-Jimenez et al. (2024), we believe that for transformer based LLMs large-scale instruction tuning further disentangles model weights, facilitating effective model merging and improving the base model's zero-shot performance.

## 4.2 Model Merging Performs Better With Bigger Models

**Experimental Setup:** In this section, we explore the effect of model size on the held-in performance of merged models. We run experiments using different model sizes, base models, merging methods, and numbers of constituent models. As in the previous experiment, we report the average results over three random seeds and compare the performance of the merged models to that of the task-specific expert models.

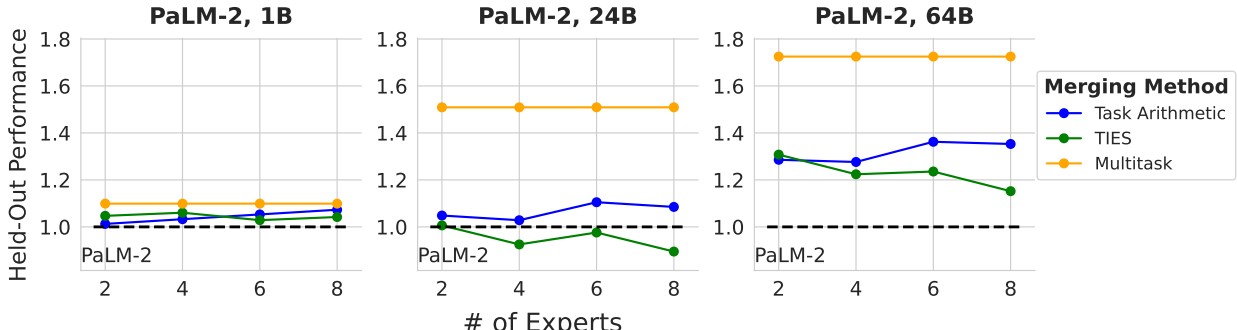

Figure 5: **Merged models at scale generalize better.** We plot the held-out generalization of the merged model for two merging methods. We also include the performance of base model (dashed line) and the multitask baseline (yellow line) which trained on a mixture of held-in tasks. We find that the number of constituent expert models (x-axis, →) had little effect on zero-shot generalization as shown in the left and center plots. However, increasing model size significantly to `64B` improved the merged model's performance over the base model (right plot). For more details see Section 4.3.

**Findings:** Figure 4 illustrates how increasing base model size impacts merging effectiveness. As model size grows (denoted by colors, ■, ■, ■, ■), merged model performance generally improves. This positive trend is consistent across all base models (different subplots), merging methods (x-axis →), and numbers of constituent models (subplots). For large instruction-tuned `PaLM-2-IT` models, the merged models perform nearly as well as task-specific expert models denoted by dashed line. These results demonstrate that for transformer based LLMs larger models facilitate merging. This suggests a promising approach for developing adaptive, modular post-training recipes. If the remaining performance gap can be further reduced, model merging could become a cost-effective alternative to multitask training. Our full results across all settings with standard deviations are available in the Appendix C.

## 4.3 Merged Models at Scale Generalize Better

**Experimental Setup:** Expert models are created by fine-tuning our base model on specialized tasks, which can lead to a decrease in its generalization capabilities. This raises the question: *How well, if at all, can the merged model generalize to held-out tasks?* Ideally, the merged model should perform at least as well as the base model on these tasks. To explore this, we evaluate the merged model's performance on unseen tasks across various model sizes, merging methods, and numbers of constituent models. Additionally, we compare these merging approach to a traditional multitask baseline, where a single model is trained on a mixture of all eight held-in task categories. As detailed in Section 3, we normalize the performance of both the merged and multitask model against the base model to assess relative gains or losses in generalization abilities.

**Findings:** Figure 2 and Figure 5 show the zero-shot generalization performance of the merged model using `PaLM-2-IT` and `PaLM-2`, respectively. Overall, we find that: ❶ The merged models outperform their corresponding base models in zero-shot generalization to held-out tasks, as indicated by performance values greater than 1 in most cases; ❷ This improvement is consistent across various model sizes (denoted by subplot), base models (different figures), merging methods (different colors ■, ■), and numbers of constituent models (on x-axis →), suggesting that merging for transformer based LLMs generally improves generalization; ❸ For weak base models (i.e., `PaLM-2`) illustrated in Figure 5, the number of constituent expert models had little effect on zero-shot generalization (Left and Center plots). However, increasing model size significantly improved the merged model's performance over the base model (Right plot); ❹ In contrast, strong base models (`PaLM-2-IT`) show a different trend, zero-shot generalization monotonically improves with the addition of more expert models as shown in Figure 2. We hypothesize this positive correlation arises as the noise learned during finetuning is canceled out when merging different expert models, resulting in better generalization; and ❺ Notably, for transformer based LLMs the merged model outperforms the multitask baseline when combining more than `6` large instruction-tuned expert models (over `24B`). This indicates that transformer

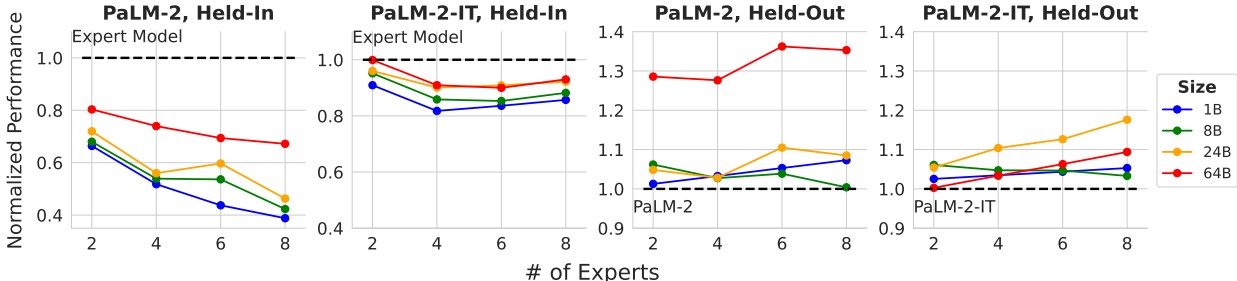

Figure 6: **Bigger model sizes can merge more experts.** We merge experts of various sizes created from `PaLM-2` and `PaLM-2-IT` models and plot the held-in (left) and held-out (right) performance of merged models. While `PaLM-2`'s held-in performance degrades with more experts, `PaLM-2-IT`'s performance stabilizes at a much higher level. Both `PaLM-2` and `PaLM-2-IT` models consistently improve held-out generalization, particularly at 24B and 64B scales with increasing expert count. For more details see Section 4.4.

based LLMs models developed through merging can generalize even better than those trained on a multitask mixture, offering a promising approach for developing highly capable language models. Our full results on other merging methods and model size are available in Appendix C.

## 4.4 Bigger Model Sizes Can Merge More Experts

**Experimental Setup:** When creating multitask models, datasets for different tasks or domains are typically combined. In contrast, model merging involves developing separate expert models for each task or domain before combining them. Previous work has shown that merging multiple models can reduce the quality of the resulting model (Yadav et al., 2024b; Ilharco et al., 2022). In this study, we experiment with merging up to 8 expert models from various base models, model sizes, and merging methods to assess their impact on successful merges.

**Findings:** Figure 6 shows the held-in and held-out performance of the merged models using Task Arithmetic as the number of constituent models increases shown on x-axis. Results for other methods can be found in Appendix C. Overall, we observe that: ❶ Unlike merging with `PaLM-2`, where held-in performance typically declines with more model merges, merging with stronger zero-shot `PaLM-2-IT` initially drops slightly in performance before stabilizing as number of constituent models increase. For example, merging eight 8B `PaLM-2` models decreases performance from 0.66 to 0.39 when increasing the number of experts from 2 to 8, whereas `PaLM-2-IT`'s performance only slightly drops from 0.91 to 0.86; ❷ In the held-out evaluations, the merged experts based on `PaLM-2` models generally outperform the base `PaLM-2` models by a small margin. However, with larger model sizes (64B), the performance improvement increases significantly, achieving about 30 percentage relative improvement. We attribute this substantial gain to the base `PaLM-2` model's weak zero-shot performance; and ❸ The merged models based on `PaLM-2-IT` show improved generalization over `PaLM-2-IT` across all settings. Additionally, for the 24B and 64B models, we observe a consistent increase in generalization capabilities with the addition of more constituent expert models.

## 4.5 Merging Llama Based Models

Given the compute constraints, we focused on a single transformers based model family PaLM-2. However, to assess the generality of some of our claims we performed experiments using the Llama-2 (Touvron et al., 2023) based models. We use existing Llama-2 based fully finetuned models, specifically we work with the WizardMath (Luo et al., 2023) and Code-Llama (Roziere et al., 2023a) models with 7B, 13B, and 70B parameters available on huggingface. We merge the Code-llama and Wizardmath models using all the 4 merging methods and present the results. The results in Table 1 show similar trends to what we observed earlier in the paper: (1) As the model size increases the performance of the merged models gets closer to the expert models on held-in tasks. (2) For large models the performance, different merging methods perform

very similar to each other. (3) Task Arithmetic, TIES, DARE-Ties typically perform better than averaging for smaller scales. Given this we believe our claims generalize to most transformer based language models.

Table 1: Merging a coding and math expert models created from Llama-2 by performing full finetuning.

| Size(↓) | Model (↓) | Un-Normalized | | | Normalized | | |
|---|---|---|---|---|---|---|---|
| | | GSM8k | MATH | HumanEval | GSM8k | MATH | HumanEval |
| **7B** | Llama-2 | 14.6 | 2.5 | 12.2 | - | - | - |
| | WizardMath | 84.1 | 43.5 | - | - | - | - |
| | Codellama-Instruct | - | - | 34.8 | - | - | - |
| | Average | 76.5 | 39.6 | 31.5 | 0.91 | 0.91 | 0.91 |
| | Task Arithmetic | 79.1 | 40.4 | 32.7 | 0.94 | 0.93 | 0.94 |
| | TIES | 80.3 | 41.2 | 33.1 | 0.95 | 0.95 | 0.95 |
| | DARE-TIES | 79.7 | 40.8 | 33.2 | 0.95 | 0.94 | 0.95 |
| **13B** | Llama-2 | 28.7 | 3.9 | 20.1 | - | - | - |
| | WizardMath | 89.7 | 50.6 | - | - | - | - |
| | Codellama-Instruct | - | - | 42.7 | - | - | - |
| | Average | 84.5 | 48.1 | 40.1 | 0.94 | 0.95 | 0.94 |
| | Task Arithmetic | 88.4 | 48.6 | 41.4 | 0.99 | 0.96 | 0.97 |
| | TIES | 87.8 | 49.4 | 41.9 | 0.98 | 0.98 | 0.98 |
| | DARE-TIES | 87.3 | 49.1 | 42.1 | 0.97 | 0.97 | 0.99 |
| **70B** | Llama-2 | 56.8 | 13.5 | 30.5 | - | - | - |
| | WizardMath | 92.8 | 58.6 | - | - | - | - |
| | Codellama-Instruct | - | - | 67.2 | - | - | - |
| | Average | 94.2 | 60.1 | 69.1 | 1.02 | 1.03 | 1.03 |
| | Task Arithmetic | 94.4 | 60.2 | 69.5 | 1.02 | 1.03 | 1.03 |
| | TIES | 94.4 | 60.3 | 69.4 | 1.02 | 1.03 | 1.03 |
| | DARE-TIES | 94.3 | 60.2 | 69.4 | 1.02 | 1.03 | 1.03 |

## 4.6 Merging Methods Become Similar at Scale

We find that all merging methods exhibit similar performance when merging large instruction-tuned transformer based models. This suggests that simpler methods like `Averaging`, can be sufficient for merging large strong expert models. Figure 7 shows the held-in and held-out performance of the 64B experts derived from `PaLM-2-IT`. All merging methods yield comparable results on both held-in and held-out tasks for any number of constituent models (shown on x-axis). We hypothesize that for transformer based LLMs as the model size increases, expert models are highly over-parameterized due to limited training data. Consequently, the subtle advantages of certain merging techniques – such as highlighting information via task vectors, resolving interference, or pruning – which benefit smaller models, become less relevant. This indicates a need for more practical and scalable methods to improve merging at scale.

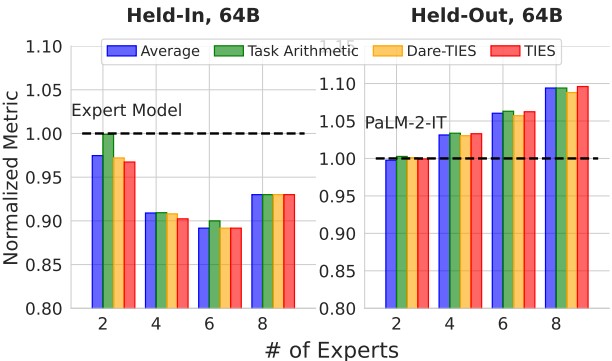

Figure 7: **Different merging methods become similar at scale.** We plot the held-in and held-out performances of merged 64B `PaLM-2-IT` models across different merging methods and numbers of constituent models. For more details see Section 4.6.

### 4.7 Discussion and Takeaways

In this section, we summarize key insights from our study and provide practical recommendations for model merging practitioners intending to use it for transformer based LLMs. Overall, we find that: ❶ Creating expert models from the best available base model is always beneficial. The quality of the base model can be gauged by its zero-shot generalization capabilities. For transformer based language models, we hypothesize that better generalization leads to improved weight disentanglement (Ortiz-Jimenez et al., 2024) and a flatter loss landscape, enhancing linear mode connectivity and facilitating model merging; ❷ Merged models often underperform compared to task-specific expert models, indicating a potential loss in performance. Despite this, specialized expert models generally outperform general-purpose multitask models (Liu et al., 2022; Roziere et al., 2023b; Luo et al., 2023), suggesting that for transformer based LLMs the performance loss may not be significant when compared to multitask models trained on specific tasks; and ❸ Our findings indicate that for `PaLM-2` models large-scale merging can accommodate more models and significantly improve generalization, outperforming multitask training when a powerful zero-shot base model is employed. ❹ Surprisingly, we find that when working with transformer based large instruction tuned models, different merging methods perform very similar. This implies that using simple merging methods like averaging will result in models that are comparable in quality with the models obtained from more advanced merging method. We hope our research inspires further fundamental studies on developing more practical and scalable merging methods.

## 5 Related Work

### 5.1 Loss Landscape and Weight Interpolation

While the loss function of a neural network is generally non-convex, recent work (Draxler et al., 2018; Freeman & Bruna, 2016; Garipov et al., 2018; Jordan et al., 2023; Gueta et al., 2023) has demonstrated that the parameter values from different training runs can sometimes be interpolated without increasing the loss (i.e. they are *mode-connected*). Many methods (Kuditipudi et al., 2019; Tatro et al., 2020; Benton et al., 2021) have explored finding these low-loss paths between models, focusing on simple (not necessarily linear) interpolations. For example, Frankle et al. (2020) showed that if a part of the optimization trajectory is shared between two neural networks then they can be interpolated without lowering accuracy. On the other hand, Neyshabur et al. (2020) showed that naively interpolating two neural networks with completely disjoint optimization trajectories can result in a catastrophic drop in their accuracies. Entezari et al. (2021) hypothesized that if we account for the permutation symmetry of neural networks, then all neural networks of a given architecture trained on the same dataset are linear mode connected. This assumption of the existence of a low-loss "basin" in parameter space encompassing the models is critical for model merging (Ilharco et al., 2023). Ainsworth et al. (2022); Singh & Jaggi (2020); Wang et al. (2020); Jordan et al. (2022); Peña et al. (2023) therefore used techniques based on finding permutations (Wang et al., 2020; Ainsworth et al., 2022) and optimal transport (Singh & Jaggi, 2020) to better align neural networks trained from scratch so that they can be merged or interpolated without increasing the loss.

### 5.2 Model Merging

Section 2.1 discusses the merging methods that we use for our experiments, however, the popularity of model merging has led to a ever-growing number of methods and applications of model merging (He et al., 2024; Daheim et al., 2023; Yadav et al., 2023a;b; 2024b; Matena & Raffel, 2022a; Jin et al., 2023). Next, we discuss some of these methods which were omitted due to large scale practical considerations. Tangent Task Arithmetic (Ortiz-Jimenez et al., 2024) fine-tune models in the tangent space for better weight disentanglement when using Task Arithmetic. Akiba et al. (2024) explore using evolutionary algorithms to choose which layers to merge. SLERP (Shoemake, 1985) and Model Stock (Jang et al., 2024) consider the geometric properties in weight space where SLERP performs spherical interpolation of model weights while Model Stock approximates a center-close weight based on several FT models, utilizing their backbone as an anchor point. Tang et al. (2023) train a mask that learns which parameters are important for the merged model. Ye et al. (2023) train a gating network to predict a weight that is then used to compute a weighted average of

examples during inference. Yadav et al. (2024a) provides a comprehensive survey of methods that train a router to route between the different models to merge. Moreover, other applications of model merging include intermediate-task training (Ramé et al., 2022b; Choshen et al., 2022a;b), continual learning (Don-Yehiya et al., 2022), model alignment (Rame et al., 2024; Ramé et al., 2024), merging pretrained models Yu et al. (2024e), or merging models in different modalities (Sung et al., 2023).

# 6    Conclusions

This study conducted a systematic, large-scale empirical investigation of model merging for transformer based language models like `PaLM-2` and `PaLM-2-IT`, addressing the limitations of previous research confined to small-scale models and limited merging scenarios. Through extensive experiments with `PaLM-2` and `PaLM-2-IT` models ranging from `1B` to `64B` parameters, we analyzed the impact of model size, base model quality, merging method, and number of experts on both in-domain and out-of-domain generalization performance. Our findings demonstrate that for these models, model merging effectively combines diverse expert knowledge particularly with increasing model size and with instruction-tuned base models. We found that larger models consistently perform better when merged and can merge more models with less performance degradation. Importantly, for transformer based LLMs model merging led to enhanced generalization capabilities, with large merged models surpassing the performance of multitask models on held-out tasks. These results show that we can develop models that generalise well in a decentralized and modular manner.

# 7    Broader Impact

We believe that this work has no broader implications of its own; however, large language models in general affects user applications and behaviors. Hence, all the implication of LLMs are also aplicable to our work.

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

# A    Detailed Task Descriptions.

We adopt the experimental setting from the T0 mixture (Sanh et al., 2021a) which contains 8 held-in and 4 held-out task categories Specifically, the 8 held-in task categories include *Multiple-choice QA* (with selected datasets DREAM (Sun et al., 2019), Cosmos QA (Huang et al., 2019)), *Extractive Qa* (Adversarial QA (Adelani et al., 2021), ROPES (Lin et al., 2019)), *Closed-Book QA* (Hotpot QA (Yang et al., 2018), Wiki QA (Yang et al., 2015)), *Sentiment Analysis* (App Reviews (), IMDB (Maas et al., 2011)), *Topic Classification* (AG News (Zhang et al., 2015), DBPedia (Lehmann et al., 2015)), *Structure-to-text* (Common Gen (Lin et al., 2020), Wiki Bio (Lebret et al., 2016)), *Summarization* (CNN Daily Mail (See et al., 2017), XSum (Narayan et al., 2018)) and *Paraphrase Identification* (MRPC (Dolan & Brockett, 2005), QQP (Iyer et al., 2017)). Similary, the 4 held-out task categories are *Sentence Completion* (with selected dataset COPA (Roemmele et al., 2011), HellaSwag (Zellers et al., 2019)), *Natural Language Inference* (ANLI (Nie et al., 2019), RTE (Dagan et al., 2005)), *Coreference Resolution* (WSC (Levesque et al., 2012b), Winogrande (Levesque et al., 2012a)) and *Word Sense Disambiguation* (WiC (Pilehvar & Camacho-Collados, 2018)).

# B    Expert Training Details

In our research, we utilized two base models, namely `PaLM-2` and `PaLM-2-IT` to create specialized expert models. We train the `PaLM-2`model for an additional 60000 steps on the Flan-v2 dataset (Longpre et al., 2023) to obtain the `PaLM-2-IT` model. We removed the T0 tasks from the flan mixture in order to training experts on them in future. Many of these training jobs were early stopped due to convergence. We used Sharded Adafactor (Shazeer & Stern, 2018) optimizer along with a cosine decay and a learning rate of 1e-4 for 1B, 24B, and 64B model sizes and 3e-5 for 8B model. We use a dropout value of 0.05. Following Chung et al. (2024), we used an input length of 2048 and output length of 512. To create expert models we perform full finetuning with the following hyperparameters. For training the experts model, for all model size, we train by default for 2000 steps with a learning rate of 3e-5 and dropout of 0.05. For some task we adjust the number of steps depending upon the convergence. For the purpose of evaluating classification tasks (Raffel et al., 2020b), we perform *rank classification*. In this method, the model's log probabilities for all potential label strings are ranked. The model's prediction is deemed accurate if the choice ranked highest aligns with the correct answer. It should be noted that rank classification evaluation can accommodate both classification tasks and multiple-choice tasks. for Task Arithmetic, TIES and DARE methods, we tested values between 0 and 1, in steps of 0.1. For TIES and DARE, we pruned 80% and 90% of the values. For TIES we pruned the bottom x% values in the task vector by magnitude, while for DARE we pruned randomly. These pruning methods are from their original papers.

# C    Full Result Tables

In this section, we provide the result for the full grid of experiments that we performed. The results contain information about any of the plots that are not provided in the main paper. Table 4 and 5 present the held-in and held-out performance of `PaLM-2` model across all model sizes, base models, merging methods, and the number of experts being merged. Similarly, Table 2 and 3 present the held-in and held-out performance of `PaLM-2-IT` model.

Table 2: The table reports the mean (std) of the normalized performance for the held-in tasks when merging experts created from `PaLM-2-IT` base models.

| Merging Method ($\downarrow$) | 1B | | | | 8B | | | | 24B | | | | 64B | | | |
|---|---|---|---|---|---|---|---|---|---|---|---|---|---|---|---|---|
| # of Experts ($\rightarrow$) | 2 | 4 | 6 | 8 | 2 | 4 | 6 | 8 | 2 | 4 | 6 | 8 | 2 | 4 | 6 | 8 |
| Average | $0.85_{(0.06)}$ | $0.78_{(0.05)}$ | $0.81_{(0.02)}$ | $0.83_{(0)}$ | $0.9_{(0.06)}$ | $0.82_{(0.05)}$ | $0.82_{(0.02)}$ | $0.85_{(0)}$ | $0.94_{(0.05)}$ | $0.84_{(0.08)}$ | $0.8_{(0.09)}$ | $0.77_{(0)}$ | $0.97_{(0.02)}$ | $0.91_{(0.01)}$ | $0.89_{(0.02)}$ | $0.93_{(0)}$ |
| Task Arithmetic | $0.91_{(0.08)}$ | $0.82_{(0.02)}$ | $0.84_{(0.01)}$ | $0.86_{(0)}$ | $0.95_{(0.06)}$ | $0.86_{(0.06)}$ | $0.85_{(0.02)}$ | $0.88_{(0)}$ | $0.96_{(0.06)}$ | $0.9_{(0.01)}$ | $0.91_{(0.01)}$ | $0.92_{(0)}$ | $1_{(0.06)}$ | $0.91_{(0.01)}$ | $0.9_{(0.01)}$ | $0.93_{(0)}$ |
| Dare-TIES | $0.9_{(0.06)}$ | $0.81_{(0.02)}$ | $0.83_{(0.02)}$ | $0.86_{(0)}$ | $0.93_{(0.03)}$ | $0.86_{(0.06)}$ | $0.84_{(0.03)}$ | $0.88_{(0)}$ | $0.94_{(0.05)}$ | $0.89_{(0.01)}$ | $0.87_{(0.03)}$ | $0.88_{(0)}$ | $0.97_{(0.02)}$ | $0.91_{(0.01)}$ | $0.89_{(0.02)}$ | $0.93_{(0)}$ |
| TIES | $0.89_{(0.05)}$ | $0.81_{(0.02)}$ | $0.82_{(0.02)}$ | $0.85_{(0)}$ | $0.93_{(0.02)}$ | $0.86_{(0.06)}$ | $0.84_{(0.03)}$ | $0.88_{(0)}$ | $0.95_{(0.04)}$ | $0.88_{(0.01)}$ | $0.86_{(0.04)}$ | $0.86_{(0)}$ | $0.97_{(0.02)}$ | $0.9_{(0.02)}$ | $0.89_{(0.02)}$ | $0.93_{(0)}$ |
| Multitask | $0.97_{(0.02)}$ | $0.96_{(0.01)}$ | $0.96_{(0.01)}$ | $0.96_{(0)}$ | $0.96_{(0.02)}$ | $0.96_{(0.01)}$ | $0.97_{(0)}$ | $0.96_{(0)}$ | $0.99_{(0.02)}$ | $0.97_{(0)}$ | $0.98_{(0)}$ | $0.98_{(0)}$ | $0.99_{(0.02)}$ | $0.98_{(0)}$ | $0.98_{(0)}$ | $0.99_{(0)}$ |

Table 3: The table reports mean (std) of the normalized performance on the held-out tasks when merging experts created from `PaLM-2-IT` base models.

| Merging Method (↓) | 1B | | | | 8B | | | | 24B | | | | 64B | | | |
|---|---|---|---|---|---|---|---|---|---|---|---|---|---|---|---|---|
| # of Experts (→) | 2 | 4 | 6 | 8 | 2 | 4 | 6 | 8 | 2 | 4 | 6 | 8 | 2 | 4 | 6 | 8 |
| Average | $0.99_{(0.04)}$ | $1_{(0.05)}$ | $1.04_{(0.02)}$ | $1.05_{(0)}$ | $1.03_{(0.02)}$ | $1.02_{(0.01)}$ | $1.03_{(0.02)}$ | $1.02_{(0)}$ | $1.05_{(0.03)}$ | $1.1_{(0.06)}$ | $1.11_{(0.08)}$ | $1.16_{(0)}$ | $1_{(0)}$ | $1.03_{(0.05)}$ | $1.06_{(0.06)}$ | $1.09_{(0)}$ |
| Task Arithmetic | $1.03_{(0.01)}$ | $1.03_{(0.02)}$ | $1.04_{(0.02)}$ | $1.05_{(0)}$ | $1.06_{(0.01)}$ | $1.05_{(0.01)}$ | $1.05_{(0.02)}$ | $1.03_{(0)}$ | $1.05_{(0.02)}$ | $1.1_{(0.06)}$ | $1.13_{(0.09)}$ | $1.18_{(0)}$ | $1_{(0)}$ | $1.03_{(0.05)}$ | $1.06_{(0.05)}$ | $1.09_{(0)}$ |
| Dare-TIES | $1.02_{(0.01)}$ | $1.03_{(0.02)}$ | $1.04_{(0.02)}$ | $1.05_{(0)}$ | $1.05_{(0.01)}$ | $1.04_{(0.01)}$ | $1.04_{(0.01)}$ | $1.03_{(0)}$ | $1.05_{(0.02)}$ | $1.1_{(0.06)}$ | $1.12_{(0.08)}$ | $1.17_{(0)}$ | $1_{(0)}$ | $1.03_{(0.05)}$ | $1.06_{(0.05)}$ | $1.09_{(0)}$ |
| TIES | $1.02_{(0.01)}$ | $1.03_{(0.02)}$ | $1.04_{(0.02)}$ | $1.05_{(0)}$ | $1.06_{(0.03)}$ | $1.05_{(0.01)}$ | $1.06_{(0.03)}$ | $1.04_{(0)}$ | $1.04_{(0.02)}$ | $1.09_{(0.06)}$ | $1.11_{(0.08)}$ | $1.16_{(0)}$ | $1_{(0)}$ | $1.03_{(0.05)}$ | $1.06_{(0.06)}$ | $1.1_{(0)}$ |
| Multitask | $1.11_{(0)}$ | $1.11_{(0)}$ | $1.11_{(0)}$ | $1.11_{(0)}$ | $1.12_{(0)}$ | $1.12_{(0)}$ | $1.12_{(0)}$ | $1.12_{(0)}$ | $1.18_{(0)}$ | $1.18_{(0)}$ | $1.18_{(0)}$ | $1.18_{(0)}$ | $1.05_{(0)}$ | $1.05_{(0)}$ | $1.05_{(0)}$ | $1.05_{(0)}$ |

Table 4: The table reports mean (std) of the normalized performance on the held-in tasks when merging experts created from `PaLM-2` base models.

| Merging Method (↓) | 1B | | | | 8B | | | | 24B | | | | 64B | | | |
|---|---|---|---|---|---|---|---|---|---|---|---|---|---|---|---|---|
| # of Experts (→) | 2 | 4 | 6 | 8 | 2 | 4 | 6 | 8 | 2 | 4 | 6 | 8 | 2 | 4 | 6 | 8 |
| Average | $0.63_{(0.09)}$ | $0.44_{(0.08)}$ | $0.36_{(0.01)}$ | $0.26_{(0)}$ | $0.66_{(0.14)}$ | $0.53_{(0.14)}$ | $0.5_{(0.13)}$ | $0.32_{(0)}$ | $0.7_{(0.18)}$ | $0.48_{(0.08)}$ | $0.51_{(0.28)}$ | $0.27_{(0)}$ | $0.8_{(0.17)}$ | $0.74_{(0.09)}$ | $0.69_{(0.06)}$ | $0.67_{(0)}$ |
| Task Arithmetic | $0.66_{(0.04)}$ | $0.52_{(0.02)}$ | $0.44_{(0.02)}$ | $0.39_{(0)}$ | $0.68_{(0.16)}$ | $0.54_{(0.14)}$ | $0.54_{(0.12)}$ | $0.42_{(0)}$ | $0.72_{(0.18)}$ | $0.56_{(0.08)}$ | $0.6_{(0.24)}$ | $0.46_{(0)}$ | $0.8_{(0.17)}$ | $0.74_{(0.09)}$ | $0.69_{(0.06)}$ | $0.67_{(0)}$ |
| Dare-TIES | $0.65_{(0.03)}$ | $0.51_{(0.03)}$ | $0.42_{(0.03)}$ | $0.37_{(0)}$ | $0.66_{(0.15)}$ | $0.51_{(0.13)}$ | $0.51_{(0.12)}$ | $0.32_{(0)}$ | $0.67_{(0.17)}$ | $0.44_{(0.02)}$ | $0.51_{(0.28)}$ | $0.27_{(0)}$ | $0.8_{(0.17)}$ | $0.74_{(0.09)}$ | $0.69_{(0.06)}$ | $0.67_{(0)}$ |
| TIES | $0.66_{(0.06)}$ | $0.5_{(0.06)}$ | $0.41_{(0.01)}$ | $0.33_{(0)}$ | $0.67_{(0.17)}$ | $0.52_{(0.16)}$ | $0.48_{(0.17)}$ | $0.29_{(0)}$ | $0.68_{(0.22)}$ | $0.49_{(0.13)}$ | $0.52_{(0.27)}$ | $0.27_{(0)}$ | $0.8_{(0.18)}$ | $0.71_{(0.11)}$ | $0.65_{(0.1)}$ | $0.56_{(0)}$ |
| Multitask | $0.88_{(0.07)}$ | $0.88_{(0.04)}$ | $0.88_{(0.03)}$ | $0.87_{(0)}$ | $1.06_{(0.09)}$ | $1.04_{(0.04)}$ | $1.04_{(0.03)}$ | $1.06_{(0)}$ | $1.25_{(0.42)}$ | $1.15_{(0.2)}$ | $1.11_{(0.14)}$ | $1.2_{(0)}$ | $0.97_{(0.03)}$ | $0.96_{(0.01)}$ | $0.96_{(0)}$ | $0.96_{(0)}$ |

Table 5: The table reports mean (std) of the normalized performance on the held-out tasks when merging experts created from `PaLM-2` base models.

| Merging Method (↓) | 1B | | | | 8B | | | | 24B | | | | 64B | | | |
|---|---|---|---|---|---|---|---|---|---|---|---|---|---|---|---|---|
| # of Experts (→) | 2 | 4 | 6 | 8 | 2 | 4 | 6 | 8 | 2 | 4 | 6 | 8 | 2 | 4 | 6 | 8 |
| Average | $0.98_{(0.03)}$ | $1_{(0.05)}$ | $1.02_{(0.03)}$ | $1.04_{(0)}$ | $1.01_{(0.12)}$ | $0.97_{(0.09)}$ | $1.02_{(0.08)}$ | $0.98_{(0)}$ | $0.95_{(0.16)}$ | $0.85_{(0.03)}$ | $0.93_{(0.18)}$ | $0.83_{(0)}$ | $1.28_{(0.14)}$ | $1.24_{(0.11)}$ | $1.29_{(0.08)}$ | $1.25_{(0)}$ |
| Task Arithmetic | $1.01_{(0.01)}$ | $1.03_{(0.04)}$ | $1.05_{(0.03)}$ | $1.07_{(0)}$ | $1.06_{(0.06)}$ | $1.03_{(0.04)}$ | $1.04_{(0.06)}$ | $1_{(0)}$ | $1.05_{(0.08)}$ | $1.03_{(0.05)}$ | $1.1_{(0.03)}$ | $1.08_{(0)}$ | $1.29_{(0.14)}$ | $1.28_{(0.13)}$ | $1.36_{(0.02)}$ | $1.35_{(0)}$ |
| Dare-TIES | $0.99_{(0.02)}$ | $1.01_{(0.05)}$ | $1.04_{(0.03)}$ | $1.05_{(0)}$ | $1.02_{(0.09)}$ | $1_{(0.06)}$ | $1.05_{(0.06)}$ | $1.01_{(0)}$ | $0.97_{(0.15)}$ | $0.89_{(0.02)}$ | $0.99_{(0.15)}$ | $0.9_{(0)}$ | $1.28_{(0.13)}$ | $1.24_{(0.11)}$ | $1.28_{(0.07)}$ | $1.24_{(0)}$ |
| TIES | $1.05_{(0.06)}$ | $1.06_{(0.05)}$ | $1.03_{(0.02)}$ | $1.04_{(0)}$ | $1.07_{(0.08)}$ | $1.04_{(0.09)}$ | $1.02_{(0.05)}$ | $0.99_{(0)}$ | $1.01_{(0.12)}$ | $0.93_{(0.04)}$ | $0.98_{(0.14)}$ | $0.9_{(0)}$ | $1.31_{(0.19)}$ | $1.22_{(0.18)}$ | $1.24_{(0.14)}$ | $1.15_{(0)}$ |
| Multitask | $1.1_{(0)}$ | $1.1_{(0)}$ | $1.1_{(0)}$ | $1.1_{(0)}$ | $1.62_{(0)}$ | $1.62_{(0)}$ | $1.62_{(0)}$ | $1.62_{(0)}$ | $1.51_{(0)}$ | $1.51_{(0)}$ | $1.51_{(0)}$ | $1.51_{(0)}$ | $1.73_{(0)}$ | $1.73_{(0)}$ | $1.73_{(0)}$ | $1.72_{(0)}$ |

