# OpenReview forum: "What Matters for Model Merging at Scale?"
_TMLR — Accepted by TMLR_

### Review · Reviewer_Lys5 · 2025-05-11

**Summary Of Contributions:**

This paper experimentally investigates the effects of scaling model size, the base model quality, and the number of expert models to be merged in model merging. In the experiments, the authors target four relatively simple merging methods and compare their performance with different model sizes and the number of expert models. The experimental result indicates that the merging expert models derived from the instruction-tuned model exhibit better performance, and larger models perform better in model merging and are not sensitive against merging methods. In addition, when merging eight large expert models, the merged models outperform the multitask-trained models. These experimental findings will provide insight for further development and practitioners of model merging.

**Audience:**

Yes

**Broader Impact Concerns:**

I do not have concerns about the ethical implications of this paper. However, it might be better to add a broader impact statement to the paper because the large-scale merged model may have an impact on various areas.

**Claims And Evidence:**

Yes

**Requested Changes:**

Some model merging methods, such as task arithmetic and TIES merging, have hyperparameters. In general, tuning such hyperparameters affects the merged model performance. How do the authors decide the hyperparameters in merging methods in the experiment? It would be better to clarify this point.

Investigating the effect and sensitivity of such hyperparameters when merging large models would be interesting. However, this extended direction is out of the scope of this paper and will be a future work.

**Strengths And Weaknesses:**

[Strengths]
- This paper provides a wide range of experiments regarding model merging and interesting insight into the effect of merge configurations.
- The experimental findings seem reasonable, and actually verifying them is valuable for the community.
- The paper is well-written and easy to understand.
- Although I am not the reviewer for the previous submission, the author's statements about the changes since the last submission are reasonable to me. Therefore, I think the paper has been improved based on the previous review.

[Weaknesses]
- As the contribution of this paper fully depends on experimental evaluation, the generalization of the result might be limited. For example, it might be difficult to generalize the result to other models than transformer-based large language models.
- The hyperparameter tuning method for the merging methods is not explained.

---

> ### Author Response · Authors · 2025-05-21
>
> We thank the reviewer for their time, effort, and for recognising the strengths of our work. We also apologize for the delay in response due to some unforeseen circumstances
>
>
> **Question-1:** As the contribution of this paper fully depends on experimental evaluation, the generalization of the result might be limited. For example, it might be difficult to generalize the result to other models than transformer-based large language models.
>
> **Response:** Thanks for pointing this out. We agree with this comment and hence at all the places in the paper we have been very careful about this and have restricted our claims to transformer based LLMs with the primary focus on Palm models.
>
> ----
>
> **Question-2:** Some model merging methods, such as task arithmetic and TIES merging, have hyperparameters. In general, tuning such hyperparameters affects the merged model performance. How do the authors decide the hyperparameters in merging methods in the experiment? It would be better to clarify this point.
>
> Investigating the effect and sensitivity of such hyperparameters when merging large models would be interesting. However, this extended direction is out of the scope of this paper and will be a future work.
>
> **Response:** As per your request, we are trying to get approvals to release all the raw data corresponding to all the hyperparameters which we can release and link in the paper.
>
> Regarding hyper parameters, for Task Arithmetic, TIES and DARE methods, we tested $\lambda$ values between 0 and 1, in steps of 0.1. For TIES and DARE, we pruned 80% and 90% of the values. For TIES we pruned the bottom x% values in the task vector by magnitude, while for DARE we pruned randomly. These pruning methods are from their original papers.
>
> ----
>
> **Question-3:** I do not have concerns about the ethical implications of this paper. However, it might be better to add a broader impact statement to the paper because the large-scale merged model may have an impact on various areas.
>
> **Response:** I have added a couple of lines of broader statements in the updated paper.
>
> **Please let us know if you have any other questions and concerns. Thanks!**

---

> > ### Comment · Reviewer_Lys5 · 2025-06-05
> >
> > Thank you for your response. I would like to ask a bit more regarding the hyperparameter setting.
> >
> > > for Task Arithmetic, TIES and DARE methods, we tested $\lambda$ values between 0 and 1, in steps of 0.1. For TIES and DARE, we pruned 80% and 90% of the values.
> >
> > What are the specific hyperparameter values used in the experimental results, such as in Figures 3 and 4?
> > I did not find the exact hyperparameter setting used for obtaining the experimental results.
> >
> > Did the authors mean that they tuned hyperparameters using a configuration or validation dataset?
> >
> > > For TIES we pruned the bottom x% values in the task vector by magnitude
> >
> > What is the exact value of "x"?

---

> > > ### Author Response · Authors · 2025-06-15
> > >
> > > **Hyperparameter setting in figure 3/4:** For each of the combination of model size, base model, # of expert models, merging methods, and seed the best hyperparamters are different and are tuned in the ranges specifed in the previous response on a validation set. Hence, there are too many setting to list the optimal hyperparameter found for each. However, overall we find that for Task Arithmetic, the optimal $\lambda$ hyperparameter for most settings was around 0.3 or 0.4, for TIES Merging and DARE optimal $\lambda$ was around 0.9/1.0 and keeping top 20% parameters was mostly optimal.
> > >
> > > **Did the authors mean that they tuned hyperparameters using a configuration or validation dataset?** Yes, this is how typically most merging methods find the optimal hyperparameters to merge the models.
> > >
> > > **What is the exact value of "x"?** The value x is a hyper parameter. For TIES and DARE, we tried x = 80% and 90%. This implies we keep top-20% or 10% parameters respectively. We found that the optimal value for most cases is to only keep the top-20% of the parameters and prune the rest.
> > >
> > > I hope this answers your questions about the hyperparameter configuration. Let me know if there are more questions.

---

> > > > ### Comment · Reviewer_Lys5 · 2025-06-16
> > > >
> > > > Thank you for the reply. I understand the hyperparameter setting, and it seems reasonable.
> > > > I have no additional questions.
> > > >
> > > > Best,

---

### Review · Reviewer_MXWs · 2025-05-17

**Summary Of Contributions:**

- This work conducts a comprehensive and systematic study on evaluating various factors influencing model merging, including model size, merging methods, the number of constituent models, and performance on held-in and held-out tasks.
- The paper presents several insightful findings derived from exhaustive empirical observations, offering valuable guidance for future model merging research.

**Audience:**

Yes

**Claims And Evidence:**

Yes

**Requested Changes:**

- Section 4.2 title is grammatically wrong, please consider to change it as “Model merging performs better with bigger model” or “Model merging becomes more effective with bigger models”.

**Strengths And Weaknesses:**

- Strengths
    - The paper is generally well-written, and the experiments are carefully designed to ensure fair comparisons.
    - It explores different merging factors and provides clear, conclusive findings for each factor, addressing gaps not adequately measured in previous works.
- Weakness
    - The authors addressed most of my requested issues. Although the Llama experiment did not perform on held-out tasks, it shows the similar trend as PaLM-2 on held-in tasks and different merging methods.

---

> ### Author Response · Authors · 2025-05-21
>
> We thank the reviewer for their time, effort, and for recognising the strengths of our work.
>
> **Question-1:** The authors addressed most of my requested issues. Although the Llama experiment did not perform on held-out tasks, it shows the similar trend as PaLM-2 on held-in tasks and different merging methods.
>
> **Response:** We are happy that we were able to address most of your concerns from the previous cycle. The Palm experiment was in a controlled setting where we ensured that during the instruction tuning phase we did not train on any of the held-out tasks to assess generalization. However, this is not possible for recent LLMs like Llama as they are very versatile and highly tuned on most of the popular tasks / benchmarks available on the internet. Hence, we did not perform the held-out evaluations. We hope that this answers your question.
>
> ----
>
> **Question-2:** Section 4.2 title is grammatically wrong, please consider to change it as “Model merging performs better with bigger model” or “Model merging becomes more effective with bigger models”.
>
> **Response:** We have fixed this in the updated paper.
>
>
> **Please let us know if you have any other questions and concerns. Thanks!**

---

> > ### Author Response · Authors · 2025-06-15
> >
> > **Please let us know if you have any other questions and concerns. Thanks!**

---

### Review · Reviewer_TVtf · 2025-05-19

**Summary Of Contributions:**

The submission undertakes an experimental analysis of model merging in LLMs across PALM and Llama-2 model families with different parameters sizes. Four model merging techniques of various complexity are considered, and models are evaluated on both held-in and held-out tasks. The submission makes several claims: model merging is more effective for base models with larger zero-shot performance; larger models facilitate "easier" merging; model merging can outperform multitask learning; and for larger models, the performance of different merging techniques is more similar.

**Audience:**

Yes

**Broader Impact Concerns:**

No broader impact concerns.

**Claims And Evidence:**

No

**Requested Changes:**

* The biggest change I would like to see would be the use of average ranks (or a similar non-parametric quantity) used as the metric for comparing multiple models on multiple datasets. Alternatively, a much stronger justification that the proposed metric is suitable in the context of this paper would also work.
* It would be nice if the standard deviations (or resulting standard errors) could be incorporated into the plots somehow. More details should be added about what precisely is different between each trial---these details could go in Appendix C with the full results tables.
* In Section 4.3 the paper refers to "our" merging approach. What is meant by this?
* Later on in Section 4.3, the phrase "model noise" is used; what is this?

**Strengths And Weaknesses:**

The submission has some strong points, in that some of the findings are quite interesting:
* The observation that stronger base models correlate well with the utility of model merging is interesting. Can the authors comment on whether there are some tasks where the base models are expected to outperform the instruction tuned models, providing an indication of whether it is the instruction tuning or improved zero-shot performance that matters?
* The finding that model merging methods all become quite similar as the size of the base model is scaled is quite interesting.
* The finding are shown to be replicated across model families; experiments are mainly conducted on the PALM fmaily of models, but some additional confirmatory experiments are run on Llama-2 models.

The main weakness is the lack of rigorous motivation for the particular normalised performance metric. There are a wealth of tools in non-parametric statistics that are designed to solve the problem the ad hoc normalisation approach in this paper attempts to solve. In particular, comparing *ranks* of methods rather than accuracies is a common approach for dealing with accuracies measured on different data distributions/datasets. The ensures that skews (and higher order moments) in the distribution of performances do not unfairly impact the comparison. In contrast, the proposed normalisation will be affected by higher order moments of the performance distribution. A very common approach used in machine learning for comparing multiple methods on multiple datasets is given by Demsar [1]. A more modern take on this is provided by Jansen et al [2]. I would strongly recommend the authors use average ranks across datasets, rather than normalised accuracies.

Some minor issues persist in this revision:
* The addition of standard deviations in the appendix is appreciated, but it is not clear what observational units are used to compute these.
* Some clarity issues, which are described below.

[1] J Demsar. "Statistical Comparisons of Classifiers over Multiple Data Sets". In JMLR, 2006.

[2] C Jansen, M Nalenz, G Schollmeyer, T Augustin. "Statistical Comparisons of Classifiers by Generalized Stochastic Dominance". In JMLR, 2023.

---

> ### Author Response · Authors · 2025-05-21
>
> We thank the reviewer for their time, effort, and for recognising the strengths of our work.
>
> ----
>
> **Question-1:** The observation that stronger base models correlate well with the utility of model merging is interesting. Can the authors comment on whether there are some tasks where the base models are expected to outperform the instruction tuned models, providing an indication of whether it is the instruction tuning or improved zero-shot performance that matters?
>
> **Response:** From our understanding between instruction tuning and zeroshot performance, zeroshot performance matters more. However, in most cases instruction tuning when done right improves the zero shot performance of the model. Good zeroshot performance means that the model has low loss on many different types of tasks that matter. Hence, it implies a flatter and well behaved loss landscape where the model connectivity property holds. Hence, the models are typically easier to merge.
>
> ----
>
> **Question-2:** The main weakness is the lack of rigorous motivation for the particular normalised performance metric. There are a wealth of tools in non-parametric statistics that are designed to solve the problem the ad-hoc normalisation approach in this paper attempts to solve.....
>
> The biggest change I would like to see would be the use of average ranks (or a similar non-parametric quantity) used as the metric for comparing multiple models on multiple datasets. Alternatively, a much stronger justification that the proposed metric is suitable in the context of this paper would also work.
>
> **Response:** Thank you for the question. We would like to point out we are not trying to “solve the problem the ad-hoc normalisation approach in this paper attempts to solve”. The normalized accuracy metric used in the paper is **not** our contribution. Moreover, it is very widely used in many very popular and seminal model merging papers like Task Arithmetic Figure-3 [6], TIES Merging Figure-5 [7], Twin-Merging Table-3 and Eqn-4 [1], Task Arithmetic in Tangent Space Table-1 [2], EMR Merging Figure-6 [3], Model Breadcrumbs Section 4.1 and Table-1 [4],  ATLAS Figure-2 [5].
>
> Moreover, in the context of model merging we only care about what fraction of the performance we are able to retain compared to some reference model (expert model on the task or the base model). Hence, the normalized accuracy is a very good metric which very clearly and simply captures this requirement by definition. We hope this answers your questions and the use of this metric.
>
> [1] Twin-Merging: Dynamic Integration of Modular Expertise in Model Merging, NeurIPS’24
>
> [2] Task Arithmetic in the Tangent Space: Improved Editing of Pre-Trained Models, NeurIPS’23
>
> [3] EMR-Merging: Tuning-Free High-Performance Model Merging, NeurIPS’24
>
> [4] Model Breadcrumbs: Scaling Multi-Task Model Merging with Sparse Masks, ECCV’24
>
> [5] Knowledge Composition using Task Vectors with Learned Anisotropic Scaling, NeurIPS’24
>
> [6] Editing Models with Task Arithmetic, ICLR’23
>
> [7] TIES-merging: resolving interference when merging models, NeurIPS’23
>
> ----
>
> **Question-3:** It would be nice if the standard deviations (or resulting standard errors) could be incorporated into the plots somehow. More details should be added about what precisely is different between each trial---these details could go in Appendix C with the full results tables.
>
> **Response:** We have already specified what seeds mean in the Section 3 “Experimental Setting”. Moreover, we have already provided an example there of what seed means – “For example, in an experiment where we merged 2 expert models, derived from the 64B Palm base model with the constituent models being MCQ and Summarization experts while the same experiment with a different seed resulted in Closed Book QA and Sentiment Analysis experts as the constituent models and so on”. Please let us know if there is still confusion about this and we will update the paper accordingly.
>
> ----
>
> **Question-4:** In Section 4.3 the paper refers to "our" merging approach. What is meant by this?
>
> **Response:** This was a typo, we fixed it in the updated paper. We meant the merging approaches used in the paper.
>
> ----
>
> **Question-5:** Later on in Section 4.3, the phrase "model noise" is used; what is this?
>
> **Response:** We have clarified this in the paper as well. “We hypothesize this positive correlation arises as the noise learned during finetuning is canceled out when merging different expert models, resulting in better generalization”
>
> ----
>
> **Please let us know if you have any other questions and concerns. Thanks!**

---

> > ### Author Response · Authors · 2025-06-15
> >
> > **Please let us know if you have any other questions and concerns. Thanks!**

---

### Decision · Action_Editor_EbLf · 2025-06-23

**Recommendation:** Accept as is

**Audience:**

Yes

**Audience Explanation:**

All reviewers from previous and this review cycle agreed that the topic of model merging, particularly its scaling properties and the factors influencing its effectiveness (model size, base model quality, number of experts), is highly relevant and interesting to the machine learning community, especially researchers and practitioners working with large language models. The paper investigates important practical questions and provides potentially useful empirical observations. These insights could guide future research directions and improve the deployment strategies of large language models.

**Claims And Evidence:**

Yes

**Claims Explanation:**

This is a resubmitted paper. The main issue raised by the previous review cycle about the generality of the authors' claims has been resolved. The paper's conclusions are considerably strengthened by (1) limiting their claims to transformer-based models and (2) including a new set of experiments on the Llama-2 model family (7B, 13B, and 70B) in Section 4.6. This shows that their main findings are not specific to the PaLM-2 architecture. Additionally, the prior lack of uncertainty quantification is effectively addressed by the inclusion of standard deviations in the appendix tables (Tables 2–5).

The authors have adequately addressed the other concerns raised by the reviewers during this review cycle. Both the reviewers and the AE are satisfied with the authors' rebuttal and the revisions made concerning the supporting evidence for all of their claims.